# Recent Advances of Diketopyrrolopyrrole Derivatives in Cancer Therapy and Imaging Applications

**DOI:** 10.3390/molecules28104097

**Published:** 2023-05-15

**Authors:** Lingyun Wang, Bihong Lai, Xueguang Ran, Hao Tang, Derong Cao

**Affiliations:** 1Key Laboratory of Functional Molecular Engineering of Guangdong Province, School of Chemistry and Chemical Engineering, South China University of Technology, 381 Wushan Road, Guangzhou 510641, China; 202221025022@mail.scut.edu.cn (B.L.); haotang@scut.edu.cn (H.T.); drcao@scut.edu.cn (D.C.); 2Institute of Animal Science, Guangdong Academy of Agricultural Sciences, Ministry of Agriculture Key Laboratory of Animal Nutrition and Feed Science in South China, State Key Laboratory of Livestock and Poultry Breeding, Guangzhou 510641, China; rxg59@aliyun.com

**Keywords:** synthesis, functionalization, diketopyrrolopyrrole (DPP), bioimaging, phototherapy, detection

## Abstract

Cancer is threatening the survival of human beings all over the world. Phototherapy (including photothermal therapy (PTT) and photodynamic therapy (PDT)) and bioimaging are important tools for imaging–mediated cancer theranostics. Diketopyrrolopyrrole (DPP) dyes have received more attention due to their high thermal and photochemical stability, efficient reactive oxygen species (ROS) generation and thermal effects, easy functionalization, and tunable photophysical properties. In this review, we outline the latest achievements of DPP derivatives in cancer therapy and imaging over the past three years. DPP-based conjugated polymers and small molecules for detection, bioimaging, PTT, photoacoustic imaging (PAI)-guided PTT, and PDT/PTT combination therapy are summarized. Their design principles and chemical structures are highlighted. The outlook, challenges, and future opportunities for the development of DPP derivatives are also presented, which will give a future perspective for cancer treatment.

## 1. Introduction

Diketopyrrolopyrrole (DPP) is readily synthesized through the reaction of an aromatic nitrile with dialkyl succinate. DPP and its derivatives were originally developed as organic pigments. Due to their advantages, such as easy synthesis and excellent stability, they have been widely used in paints, inks, organic solar cells [1], fluorescent sensors [2], and so on. As a kind of functional material, DPP possesses distinct characteristics such as facile structural modifications, efficient reactive oxygen species (ROS) generation and thermal effects, tunable absorption and emission properties, and high thermal stability. DPP-based fluorescent probes for various analytes have been developed [3]. Because of their easy internalization into mammalian cells, DPP dyes also exhibit outstanding advantages for imaging applications such as molecular probes. Moreover, due to their different properties, DPP derivatives have shown excellent PDT and PTT performance in cancer treatment (Figure 1).

Cancer has threatened human health in the world. The current cancer treatment methods mainly include radiotherapy, chemotherapy, and surgical resection. However, such traditional treatment methods often cause a series of side effects. Therefore, a safe and effective new tumor treatment has become an urgent problem to be solved. Phototherapy, mainly including photothermal therapy (PTT) and photodynamic therapy (PDT), shows promising anti-tumor performance. PTT works primarily by converting light into heat, causing local high temperatures in diseased tissues, and ablating tumors through apoptosis and necrosis or protein degeneration of tumor cells. In addition, PDT exerts its action primarily by irradiating the tumor site with a specific wavelength, which can activate the photochemical reaction to destroy tumor cells. Photosensitizing drugs transfer energy to surrounding oxygen, producing highly active singlet oxygen (^1^O_2_). Singlet oxygen can oxidize nearby biological macromolecules, produce cytotoxicity, and kill tumor cells [4,5,6,7]. Phototherapy is a potential therapeutic modality for cancer treatment because of its noninvasiveness, reduced damage to normal tissues, fewer side effects, and high selectivity compared with surgery, chemotherapy, and radiotherapy. Extensive research is being carried out to develop DPP-based photosensitizers for malignant tumor treatment [8,9,10,11]. As we know, the PDT effect is weakened due to hypoxia in cancer treatment. PTT is an effective therapy that can overcome hypoxia in tumors, as it is oxygen-independent and has high spatiotemporal accuracy. Moreover, for PDT, the traditional photosensitizers are limited in clinical use due to photobleaching, photodegradation, and dark toxicity. Imaging-guided combination therapy would be an effective strategy for cancer treatment to overcome these shortcomings. In this review, we outline the latest achievements of DPP derivatives in cancer therapy and imaging over the past three years. DPP-based conjugated polymers and small molecules for detection, bioimaging, PTT, photoacoustic imaging (PAI)-guided PTT, and PDT/PTT combination therapy are summarized. We believe that this review will give a future perspective for the development of high-performance DPP dyes for bioimaging and phototherapeutic applications.

## 2. Application

### 2.1. Imaging and Detection

Fluorescence imaging is an effective approach for tracking the biological processes of cancer and disease due to its high sensitivity, high signal-to-noise ratio, real-time monitoring, and no sample damage. The development of DPP-based biomaterials is of great significance for the real-time monitoring of disease development [12,13,14]. For example, bis-phosphonate functionalized DPP was used as a fluorescent probe for in vitro bone imaging [13]. DPP-based fluorescence probes for the imaging of lysosomal Zn^2+^ and identification of prostate cancer in human tissue were reported [15]. Most mitochondria-targeting fluorescent molecules have triphenylphosphonium and pyridinium cations. Recently, a neutral phosphine oxide DPP compound (**PhODPP**), which can preferentially aggregate at the mitochondria at nanomolar concentrations, was developed [16]. **PhODPP** showed comparable performance to the commercially available Mitotracker Red staining agent (Figure 2a). Its fluorescence quantum yield significantly increased by 7 times. **PhODPP** was the first uncharged DPP for the selective imaging of mitochondria. Another DPP derivative (**T25**) based on nanoparticles with NIR-IIa emission was utilized for fluorescence angiography and cerebral vascular microscopic imaging (Figure 2b) [17]. An 800 μm penetrating depth and excellent signal–background ratios of 4.07 and 2.26 (at 250 and 400 μm, respectively) were achieved. Because it produced a high quantum yield of 1.84% in the range 800–1400 nm and a high-spatial-resolution of 3.84, accurate observation of small metastatic tumors (0.3 mm × 1.0 mm) could be achieved through NIR-IIa fluorescence imaging with high spatial resolution and position.

A mannose-substituted AIE-active DPP probe was developed for lectin detection [18]. This was the first fluorogenic AIE-based probe that could sense lectins in the NIR region, but it suffered from poor water solubility and failure in the fluorescence imaging of cancer cells. Recently, another AIE-active mannosylated-DPP (**DPPS-M**) containing six mannose groups was synthesized to expand the application in the imaging of cancer cells [19]. Through sugar–lectin interactions to form aggregates, the fluorescence of **DPPS-M** increased after the addition of lectin. **DPPS-M** selectively recognized concanavalin A (Con A) with fluorescence enhancement and nanomolar limits of detection (Figure 2c). Due to its improved water solubility, **DPPS-M** could be used for the fluorescence imaging of cancer cells. In addition, through the use of an electrostatic-interaction-driven assembly, a cationic DPP amphiphile was applied as a turn-on fluorescent probe for bovine serum albumin, with a limit of detection of 0.08 μmol/L [20]. The excellent cell compatibility was successfully applied to light up BSA-rich HeLa cells.

**Figure 2 molecules-28-04097-f002:**
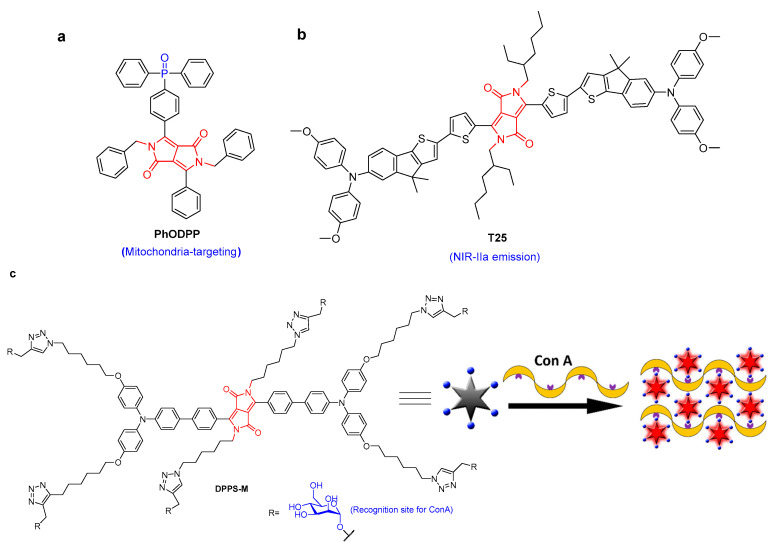
The structure of (**a**) **PhDPP** and (**b**) **T25**. (**c**) **DPPS-M** for Con A detection based on the AIE mechanism [19] {Adapted with permission from Ref. [19]. 2019, Wang, J et al.}.

Excessive superoxide anions can stimulate autophagy or apoptosis signals in cells, leading to cell death and causing various diseases. When DPP was linked with diphenyl phosphinate at the side chain as a reaction site of superoxide anions, the resulting **DPP-S** showed high sensitivity to superoxide anions, yielding an emission shift from 652 to 545 nm and a low limit of detection of 20.5 nM (Figure 3a) [21]. Due to the presence of pyridine cations, **DPP-S** could target mitochondria and could be employed as a ratiometric fluorescent probe for endogenous superoxide anion detection in MCF-7 and RAW264.7 cells and in vivo experiments. In addition, the up-regulation of peroxynitrite (ONOO^−^) levels in the liver are relative to acetaminophen (APAP)-induced liver injury. Two DPP-based ratiometric fluorescent probes (**DPP-DH-P** and **DPP-DEG-P**) for detecting and imaging peroxynitrite were reported (Figure 3b) [22]. **DPP-DH-P** exhibited a higher signal-to-noise ratio (2750-fold) and a lower detection limit (3.5 nM) for tracking ONOO^−^ in solution than **DPP-DEG-P**. However, **DPP-DEG-P** with a hydrophilic chain showed better biocompatibility, and could monitor the fluctuation of ONOO^−^ in APAP-treated hepatocytes with a high signal-to-noise ratio (20-fold) via ratiometric fluorescence imaging.

Using the fluorescence resonance energy transfer (FRET) and the aggregation-induced emission (AIE) strategies, Nie et al. developed an organic micellar nanoprobe via the co-assembly of phenyl-diketopyrrolopyrrole (**PDPP1**) and amphiphilic tetraphenylethene (**TPE1**) in aqueous solution for the ratiometric detection of hypochlorite (Figure 4) [23]. Upon addition of hypochlorite, C=C bonds in the DPP core were oxidized and the FRET process between **PDPP1**/**TPE1** was broken, resulting in the enhanced emission of **TPE1** and decayed emission of **PDPP1**. The low sensitivity with LOD (32 nM) in a ratiometric manner and the imaging of exogenous and endogenous hypochlorite in live cells were achieved.

### 2.2. Detection and PDT

Typical photosensitizers can cause phototoxicity in both normal cells and cancer tissue. The activatable photosensitizers (aPSs) in PDT exhibit improved selectivity and sensitivity towards cancer cells. Hua’s group developed some aPSs for imaging-guided PDT. For instance, **DPP-GGT** showed highly selective and obvious fluorescent changes from red to yellow for γ-glutamyltranspeptidase (γ-GT, tumor-related biomarker) detection (Figure 5a) [24]. More importantly, the photosensitizing ability of **DPP-GGT** was triggered in the presence of γ-GT. Based on this process, **DPP-GGT** not only specifically detected endogenous γ-GT in liver tumors, but was also used as a γ-GT triggered photosensitizer. The resulting product exhibited photodynamic killing effects on human hepatic cancer cells (HepG2) via overexpressed γ-GT in tumors. **DPP-BPYS** showed a redshifted spectrum and AIE activity at the same time. Based on the AIE mechanism, the H_2_O_2_-activated **DPP-BPYS** showed “turn on” near-infrared fluorescence emission, high singlet oxygen (^1^O_2_)-generating capacity, and targeting of lipid droplets (LDs) in tumor cells (Figure 5b) [25]. The rapid imaging of endogenous H_2_O_2_ in vivo was demonstrated. Phototoxicity experiments have shown that **DPP-BPYS** effectively ablates tumor cells by inducing cell apoptosis under H_2_O_2_ and white-light irradiation. In another case, **DPP-Leu** was developed for leucine aminopeptidase (LAP) detection based on the specific enzymatic cleavage of the N-terminal leucine residue, leading to distinguishing tumor cells with a high LAP content from normal cells (Figure 5c) [26]. A detection limit of 0.011 U·L^−1^ was obtained, which could effectively quantitatively detect LAP in fetal bovine serum (FBS) and artificial urine. Cell-imaging experiments have shown that **DPP-Leu** can target mitochondria and distinguish tumor cells with different LAP levels. At the same time, activatable **DPP-Leu** can generate ROS to kill tumor cells under light irradiation without damaging normal cells. Thienyl-substituted diketopyrrolopyrrole (**SDPP-DM**) with an α, β-unsaturated double-bond as a recognition site for the detection of endogenous bisulfite was reported, yielding obvious color changes from green to pink and fluorescence enhancement due to the reduced conjugated length (Figure 5d) [27]. In addition, **SDPP-DM** has been successfully applied to imaging and distinguishing different endogenous bisulfite levels in normal liver cells and cancer cells. **SDPP-DM** has shown ROS generation and phototoxicity triggered by endogenous bisulfite, resulting in a potential application for liver-cancer diagnosis and HepG2 cell killing.

Dong’s group synthesized glutathione (GSH)-responsive DPP derivatives (**DPPBPh** and **DPPTPh**) (Figure 6) [28]. High concentrations of GSH could reduce the damage of ROS to cancer cells, severely diminishing the efficacy of PDT in cancer treatment. In two cases, the thiol group in GSH reacted with the –CN group to form thiazole through the Michael addition reaction and recovered the quenched fluorescence. **DPPTPh** NPs possess higher ^1^O_2_ efficiency and photothermal conversion efficiency than **DPPBPh** NPs. In vitro and in vivo experiments have shown that **DPPTPh** can accumulate at tumor sites. Under laser irradiation, tumor growth was significantly inhibited, while other normal tissues were not damaged.

### 2.3. PTT and Photoacoustic Imaging (PAI)-Guided PTT

A conjugated polymer based on porphyrin and DPP (**Ppordpp**) showed efficient absorption extending in NIR-II and high photothermal conservation efficiency (PCE) of 86.21% (Figure 7a) [29]. Under radiation, **Ppordpp** NPs transferred energy in the form of heat to kill cancer cells. Similarly, another DPP-based conjugated polymer (**PDPP-TP**) had a PCE of 52.8%, leading to efficient photothermal antibacterial treatment with almost 100% efficiency against Gram-negative *E. coli* and Gram-positive *S. aureus* (Figure 7b) [30]. Zheng et al. developed three amphiphilic DPP derivatives (**TPADPP, DTPADPP,** and **TPADDPP**) with different poly-(ethylene glycol) side chains (Figure 7c). The corresponding nanoparticles (NPs) were obtained via self-assembly. The photothermal conversion efficiencies of **DTPADPP** NPs and **TPADDPP** NPs were 48.1% and 41.7%, respectively. **DTPADPP** NPs and **TPADDPP** NPs can significantly accumulate in tumor tissue for real-time in vivo fluorescence imaging. In vivo experiments have shown that **DTPADPP** NPs and **TPADDPP** NPs have photothermal effects and efficient tumor ablation ability [31].

As we know, nanoagents with efficient NIR-absorbing capacity are ideal probes for photoacoustic imaging (PAI). PAI-guided PTT shows promising applications in the biomedical field [32,33]. For example, porphyrin derivatives have advantages such as high extinction coefficients, good biocompatibility, and low side effects, but they always lack photostability in the NIR region. Due to their good photostability and other characteristics, DPP derivatives can serve as electron acceptors to form D-A structures with porphyrins, making up for their shortcomings. Based on this mechanism, the porphyrin-DPP conjugate (**Por-DPP**) exhibited a PCE of 62.5% and a distinct PTT effect under 808 nm laser irradiation (Figure 8a) [34]. The PAI signal intensity of the tumor region was 13 at 8 h, which was 4.3-fold higher than that of the background of the tumor, indicating that **Por-DPP** NPs could passively target tumor tissue due to their efficient EPR effect. The significant therapeutic effect of **Por-DPP** NPs under laser irradiation was further confirmed by evaluating the PTT effect in vivo through using a HeLa tumor-bearing mouse model. An IC_50_ value of 11.6 μg/mL with no significant side effects after phototherapy was achieved.

Jin et al. investigated the effects of heteroatom substitution in DPP polymers for PAI/PTT cancer ablation (Figure 8b) [35]. The substitution of heteroatoms and changes from O to S and then to Se of DPP conjugated polymers could significantly regulate the absorption spectra and energy gap. The PCE value and absorption coefficient of **DPP-SO** NPs were much higher than those of **DPP-SS** and **DPP-SSe** NPs under 808 nm irradiation, and **DPP-SO** NPs exhibited significant PA signals. Remarkably, the IC_50_ value of **DPP-SO** for killing A549 cells was half that of **DPP-SS** and **DPP-SSe** NPs. Moreover, due to the EPR effect, **DPP-SO** NPs could accumulate at the tumor site. An in vivo experiment indicated that cancer cells could be killed by necrosis and apoptosis under laser irradiation without causing damage to other parts.

Intra- and intermolecular interactions were studied with DPP derivatives containing chalcogen and fluorine atoms (Figure 8c) [36]. The synergistic π–π and F–H interactions facilitated fluorine- and selenium-substituted **DPP-SeF** with the highest PCE of 62% (32% for **DPP-SS**). The IC_50_ value of ∼8.36 μg·mL^−1^ for **DPP-SeF** was lower than that of 15.14 μg·mL^−1^ for **DPP-SS** on A549 cells under 808 nm light irradiation.

### 2.4. PDT/PTT Combination Therapy

DPP-based conjugated polymers have good photostability and large molar extinction coefficients [37,38,39]. Polyphenylenevinylene (PPV) and DPP-conjugated polymers were developed to construct PTT and PDT dual-mode NPs for antibacterial application [40]. Liu et al. synthesized the DPP-based conjugated polymer **DBT** for PDT/PTT combination therapy (Figure 9a) [41]. **DBT** NPs showed a high mass-extinction coefficient of 5.407 cm^−1^ mg^−1^ mL at 808 nm, moderate singlet oxygen yield, photothermal conversion efficiency (41.5%), and second near-infrared window (NIR-II) emission at 1056 nm with a quantum yield (QY) of 0.16%. In vitro and in vivo experiments indicated that **DBT** NPs show high-performance PTT/PDT combination therapy. In order to increase the light absorption coefficient, a D-A-D-type conjugated polymer (**PBDPP**) was synthesized (Figure 9b) [42]. **PBDPP** NPs showed PCE up to 60% due to the low energy loss of radiative transitions generated by the D-A-D structure and aggregation-induced quenching in nanoparticles. More importantly, **PBDPP** NPs have exhibited precise glioblastoma-specific capability and can effectively kill glioblastoma cells both in vitro and in vivo. Under 808 nm irradiation, **PBDPP** NPs induced remarkable cell death with an IC_50_ of 0.15 μg·mL^−1^ and complete tumor elimination using a 0.35 mg.mL^−1^ dosage in an in vivo mouse experiment without any side effects.

Hyaluronic acid (HA)-functionalized DPP (**HA-Cys-TTDTT**) and chlorin e6 (Ce6) were encapsulated by self-assembly for tumor-targeting and multimodal imaging-guided PDT/PTT synergistic therapy (Figure 9c) [43]. The resulting NPs showed ^1^O_2_ quantum yields (53.0%), enhanced fluorescence intensity due to an efficient energy transfer from DPP dye to Ce6, and a moderate PCE (*η* = 37.7%) under laser irradiation at 635 nm. NIR fluorescence and thermal imaging indicated that **HA-Cys-TTDTT** NPs were located at the tumor sites by the HA active targeting, producing enhanced cytotoxicity to tumor cells. In addition, Ce6@**HA-Cys-TTDTT** NPs could degrade gradually in tumor cells through excessive expression of HA.

Mao’s group developed a hybrid platform based on Ir nanoparticles and a DPP-conjugated polymer [44]. Hybrid nanoparticles produced oxygen in the presence of endogenous H_2_O_2_ for down-regulation of hypoxia-inducible factor 1 subunit α (HIF-1α) protein, thereby reversing the tumor hypoxia microenvironment. In this case, Ir nanoparticles were used as PTT agents and nanocatalysts to generate oxygen, resulting in enhanced PTT effects in the hybrid platform. The PCE value under 808 nm irradiation was up to 67.0%. By using the 4T1 tumor-bearing mouse model to investigate the anticancer activity of **DPP-Ir** NPs, excellent PTT anti-tumor efficacy, low biological dark toxicity, compatibility, and blood compatibility were achieved. Tang’s group proposed a strategy based on to fabricate NIR-II dyes with both high absorbance and excellent signal outputs (fluorescence and heat). Conjugated fluorophores **TADAT** and **TDADT** with one and two DPP units, respectively, were synthesized (Figure 10b,c) [45]. Two highly twisted triphenylamine (TPA) and alkylthiophene–benzobisthiadiazole–alkylthiophene moieties were introduced to prevent intermolecular π–π interactions. **TADAT** and **TDADT** exhibited fluorescence quantum yields of 0.2% and 0.1% and PCE values of 64.3% and 60.4%, respectively. **TDADA** NPs provided excellent high-resolution imaging to improve the accuracy of cancer surgery.

In recent years, semiconductor materials have been widely used in biomedical applications, including imaging and treatment. The 2-pyridone group is able to react with singlet oxygen to form endoperoxides of 2-pyridone, which can undergo thermal cycloreversion to release singlet oxygen in vivo, re-producing the 2-pyridone [46,47]. In order to continuously deliver singlet oxygen in the dark and hypoxic tumor microenvironment, an amphiphilic polymer containing a 2-pyridone unit (PEG-Py) was used to encapsulate **DPPTPE** to produce smart phototheranostics (Figure 10e) [48]. **DPPTPE**@PEG-Py NPs kept producing ^1^O_2_ under laser irradiation even when the O_2_ supply was stopped. The singlet oxygen yield of the NPs was 69% and the PCE was 30.6%. Moreover, due to the large Stokes shift of **DPPTPE**, in vivo fluorescence-imaging-guided PTT/PDT combination therapy was obtained by inhibiting tumor growth without side effects on major organs.

Li et al. designed and synthesized a mitochondria-targeting **DPP2+** by introducing imidazole groups (Figure 10d) for synergistic PDT/PTT, which could produce PCE of 35% and singlet oxygen under 635 nm laser irradiation [49]. Importantly, **DPP2+** NPs exhibited enhanced cell uptake, specific mitochondria-targeting ability, significant inhibitory effects, and low side effects on tumors through PTT/PDT synergistic effects.

Through the use of an acceptor planarization and donor rotation strategy, 3,6-divinylsubstituted diketopyrrolopyrrole (DPP) derivatives (**2TPEVDPP** with four rotors) as type-I PSs were synthesized (Figure 10h) [50]. By introducing a vinyl linker, the flatness and the π-conjugation of the compound were improved. This helped to enhance the intersystem crossing (ISC) and the production capacity of ROS, further promoting the redshift of the absorption wavelength. **2TPEVDPP** NPs keep a good balance between ROS generation and the heat dissipation pathway (PCE = 66%). Under irradiation, **2TPEVDPP** NPs could effectively inhibit tumor growth without side effects. **2TPEVDPP NPs** showed potential application for in vivo NIR fluorescence-imaging-guided synergistic PDT/PTT therapy. For a donor–acceptor compound, **PDBr** NPs (Figure 10f) with a high singlet oxygen (^1^O_2_) quantum yield of 67%, PCE of 35.7%, and excellent fluorescence/infrared-thermal imaging performance were developed [51]. **PDBr** could significantly inhibit the growth of living mouse tumors by combining PDT/PTT with the help of imaging guidance. Recently, through the use of extended conjugation and enhanced TICT effects, **TPA-TDPP** NPs for NIR fluorescence-image-guided PDT/PTT were reported (Figure 10g) [52]. A ^1^O_2_ production capacity of 50% and PCE of 38.7% were shown. **TPA-TDPP** NPs can accumulate at tumor sites through the EPR effect, and inhibit tumor growth through the synergistic effect of PTT/PDT.

Black phosphorus nanosheets (BPNs) largely possess a specific surface area, excellent photothermal conversion efficacy (28.7%), and negligible dark toxicity. Recently, Li et al. reported an A-D-A-type DPP photosensitizer (**AN(DPP)_2_**) (Figure 10a), which was loaded onto PEGylated BPNs [53]. The resulting NPs showed a two-dimensional planar morphology with lateral sizes of 190 nm and an average thickness of 3.3 nm. The loading and encapsulation efficiency of **AN(DPP)_2_** were 5.8% and 96.7%, respectively. Moreover, a PCE of 29.1% and ^1^O_2_ generation capacity of 89.8% were found. As a result, a remarkable antitumor effect toward 4T1 cells and metastatic breast cancer was shown upon light irradiation.

**Figure 10 molecules-28-04097-f010:**
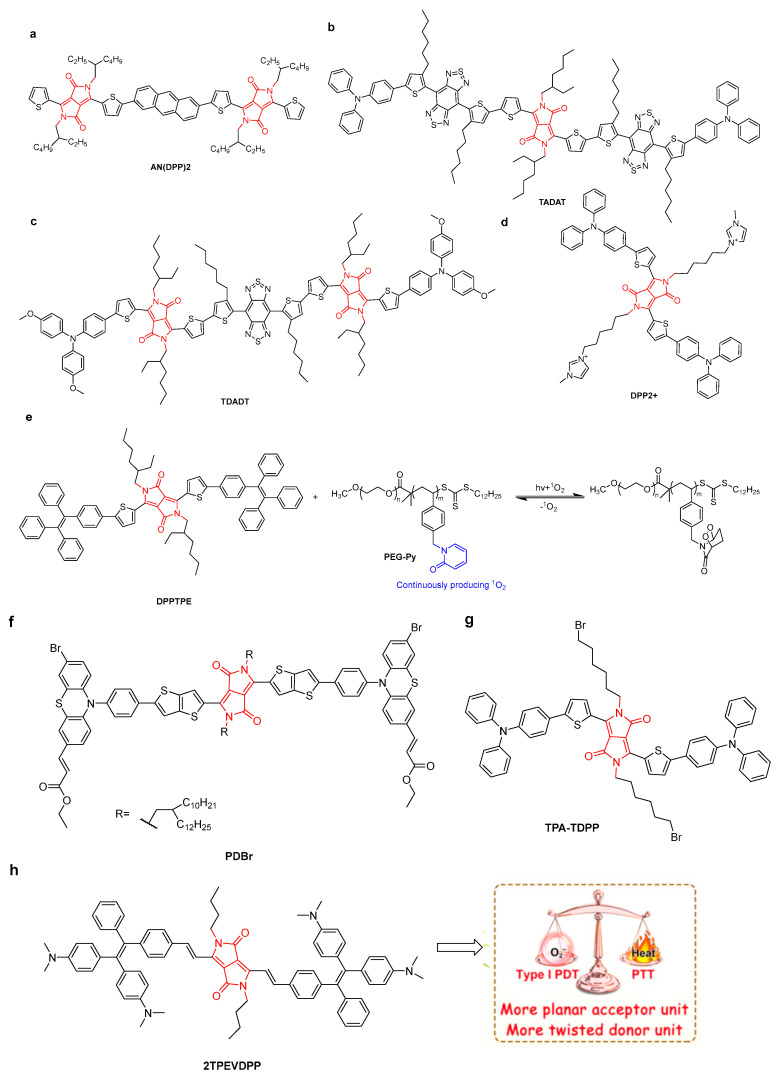
The structure of (**a**) **AN(DPP)_2_**, (**b**) **TADAT**, (**c**) **TDADT**, (**d**) **DPP_2_^+^**], (**f**) **PDBr** [51], and (**g**) **TPA-TDPP** [52]. (**e**) Schematic of light-responsive PEG-Py with ^1^O_2_ capture-and-release ability for phototheranostics. (**h**) The structure of **2TPEVDPP** and application as NIR fluorescence-imaging-guided synergistic type-I PDT-PTT cancer theranostics {Adapted with permission from Ref. [50]. 2022, Feng, L et al.}.

## 3. Conclusions

In this review, we summarized the recent progress of DPP dyes and their applications in cancer diagnosis and treatment [54,55,56,57,58,59]. Their applications, including bioimaging, detection, PDT, PTT, and PAI, have shown enormous potential due to their superior photophysical properties and phototheranostics character.

Phototherapy has great advantages in increasing the effect of tumor treatment and reducing side effects. Efforts have been made to design a series of clinical applications of cancer therapy. However, some limitations cannot be ignored. Firstly, traditional PDT is highly dependent on oxygen, but the tumor microenvironment is highly hypoxic. Most DPP derivatives are highly O_2_-dependent type II PSs (generation of singlet oxygen), which greatly diminishes the anticancer outcomes. Second, many DPP-based phototheranostics are passively accumulated in the tumor tissues through the EPR effect, leading to harm to normal tissue and long post-treatment darkroom processing time. Third, DPP-based fluorescent materials mainly work in the NIR-I biological window for bioimaging, which limits application in deep tissues and tumors. Moreover, weak emission signals have been shown due to the aggregation-caused quenching (ACQ) effect. Fourth, the structural properties and applications of DPP derivatives should be clarified, which is important to understand and address the limiting factors for DPP-based phototheranostics. Fifth, the drawbacks of phototherapy agents include poor water solubility due to hydrophobicity. For biological application, the in vivo degradation of DPP derivatives should also be considered and evaluated.

From our perspective, the following suggestions should be considered in the future in the development of DPP derivatives in biological applications.

Both efficient ROS production efficiency and good photothermal conversion efficiency can be achieved via collaborative processing of PDT/PTT. Moreover, since immunotherapy is a promising cancer treatment approach, DPP-based dyes for combinational phototherapy and immunotherapy have been developed [60]. Moreover, multifunctional DPP materials created by combining optical/acoustic/magnetic imaging modes with other therapeutic modes (chemodynamic therapy or gene therapy, etc.) are promising in practical applications.NIR-II fluorescence imaging, as a non-invasive imaging technology that provides centimeter-level depth and micron-level resolution, has been investigated. In order to further broaden the application of DPP derivatives, it is necessary to further develop DPP derivatives that can be applied to NIR-II fluorescence bioimaging.The practical clinical application is limited by issues such as biocompatibility, cytotoxicity, targeting specificity, and biodegradability. For example, the mitochondria of normal cells and cancer cells are different, which has led to the development of DPP derivatives that can target mitochondria to improve the effectiveness of treatment. The targeting ability and response to external stimuli are also important during molecular design.NIR-II image-guided synergistic enhancement for the in vivo chemo-photodynamic therapy of osteosarcoma has been developed [61]. DPP was used as a fluorescent and gene-loading capacity vector for drug delivery and tumor imaging in vitro and in vivo [62].Cancer cell membranes can be penetrated by nanomaterials, allowing accumulation in diseased areas and improvement of treatment effectiveness. Converting DPP derivatives into nanoreagents should be considered.A simpler synthesis route of DPP derivatives should be designed to achieve maximum effects at the lowest cost.

## Figures and Tables

**Figure 1 molecules-28-04097-f001:**
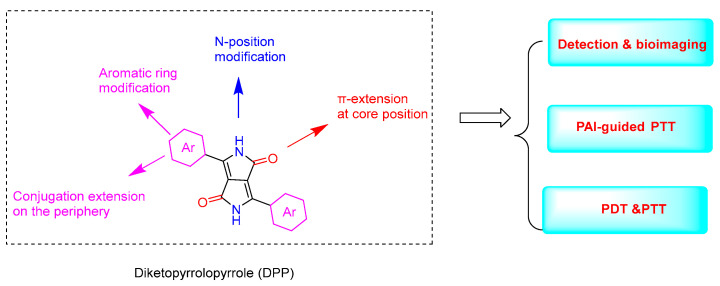
The structure of DPP and its derivatizable sites and their applications.

**Figure 3 molecules-28-04097-f003:**
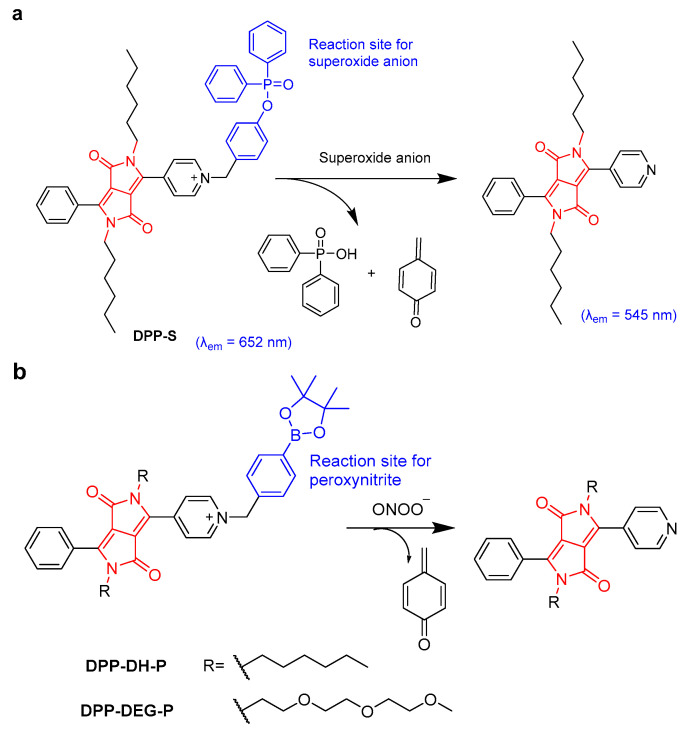
(**a**) The proposed mechanism between **DPP-S** and superoxide anion. (**b**) The proposed mechanism between **DPP-DH-P**, **DPP-DEG-P**, and ONOO^−^.

**Figure 4 molecules-28-04097-f004:**
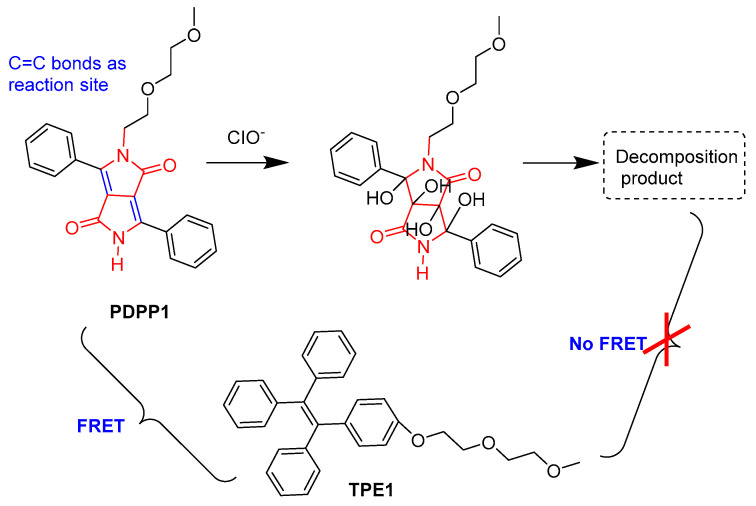
The detection of endogenous ClO^−^ with **TPE1/PDPP1** micellar nanoprobe.

**Figure 5 molecules-28-04097-f005:**
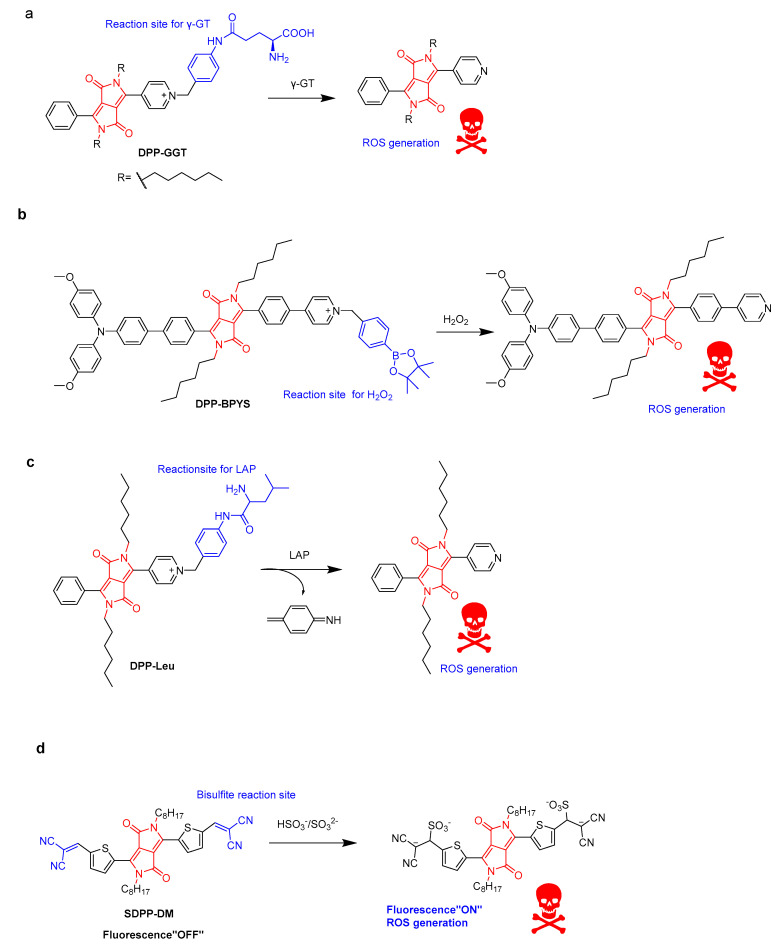
Schematic illustration of (**a**) γ-GT-triggered **DPP-GGT**, (**b**) H_2_O_2_-triggered **DPP-BPYS**, (**c**) LAP-triggered **DPP-Leu**, and (**d**) bisulfite-triggered **SDPP-DM**.

**Figure 6 molecules-28-04097-f006:**
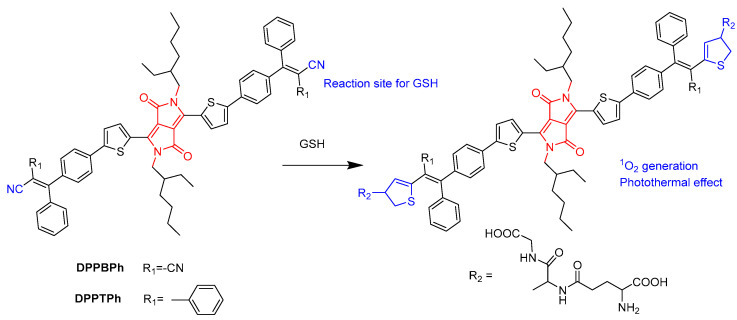
Structure of **DPPBPh** and illustration of GSH-responsive DPP derivatives.

**Figure 7 molecules-28-04097-f007:**
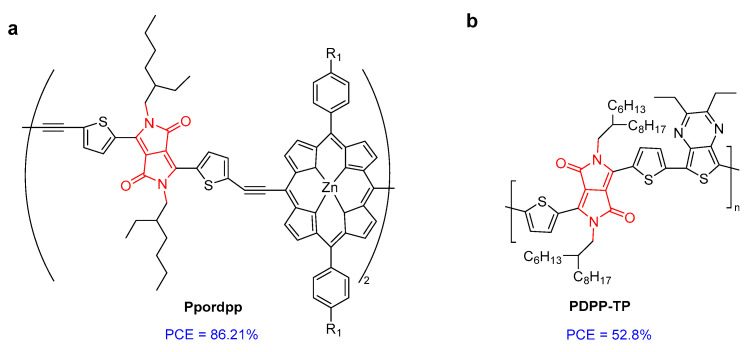
The structure of (**a**) **Ppordpp**; (**b**) **PDPP-TP**; and (**c**) **TPADPP-PEG**, **DTPADPP-PEG**, and **TPADDPP-PEG**.

**Figure 8 molecules-28-04097-f008:**
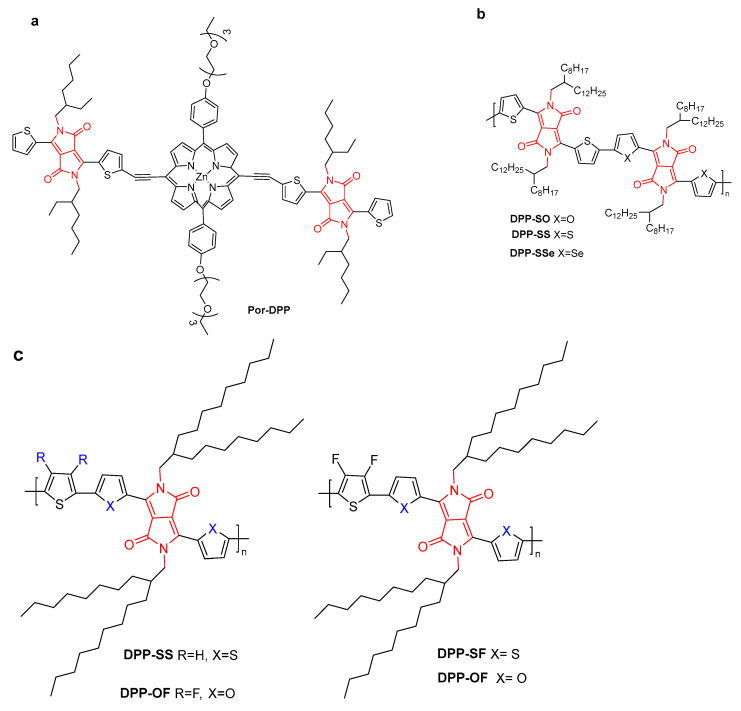
The structure of (**a**) **Por-DPP**; (**b**) **DPP-SO**, **DPP-SS**, and **DPP-SSe**; and (**c**) **DPP-SS**, **DPP-OF**, **DPP-SF** and **DPP-OF**.

**Figure 9 molecules-28-04097-f009:**
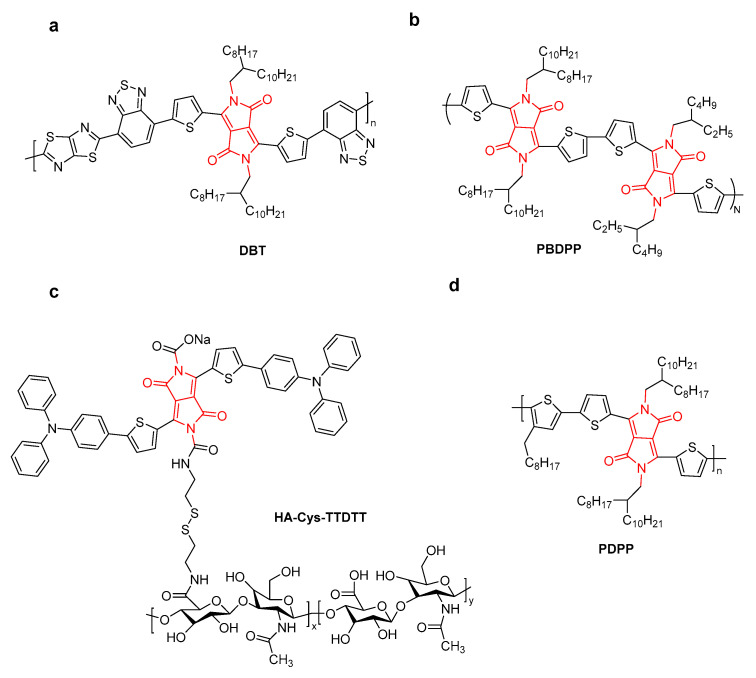
The structure of (**a**) **DBT**, (**b**) **PBDPP**, (**c**) **HA-Cys-TTDTT**, and (**d**) **PDPP**.

## Data Availability

Not applicable.

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
