# Peer review of "Recent Advances of Diketopyrrolopyrrole Derivatives in Cancer Therapy and Imaging Applications"

_molecules, 2023, doi:10.3390/molecules28104097_

Round 1

Reviewer 1 Report

The manuscript "Recent Advances of Diketopyrrolopyrrole Derivatives in Cancer Therapy and Imaging Applications " discuss the importance of phototherapy and bioimaging in cancer treatment, and highlight the unique properties of DPP dyes that make them effective for these purposes, including high thermal and photochemical stability, efficient reactive oxygen species (ROS) generation, and tunable photophysical properties. The review covers the latest research on DPP derivatives in cancer therapy and imaging over the past 6 years, including studies on their use in photothermal therapy (PTT) and photodynamic therapy (PDT), as well as their potential future applications.

The manuscript is well prepared in terms of figures and broadness of information covered hence the review only covers research on DPP derivatives in cancer therapy and imaging over the past 6 years with very limited geographical orgin, which may not provide a comprehensive overview of all relevant studies in this field. The review focuses primarily on the properties and potential applications of DPP derivatives, but may not provide a detailed analysis of their limitations or challenges in clinical settings.

The other weakness is the level of English language. There are numerous incorrect sentences but also the overall readability of the text is rather poor and its logical structure should be improved before publication.

I suggest adding citations to literature sorces in all figure captions.

Here are some suggested changes:

frequent errors, such as missing articles and prepositions.

p2l40 should read : "Cancer has become one of the main diseases that endanger human health in the world today. A safe and effective new type of cancer treatment has become an urgent problem to be solved."

p2.l.65 is this the intendend mening? "Fluorescence imaging is an effective approach for tracking biological processes in cancer and diseases, due to its high sensitivity, high signal-to-noise ratio, real-time monitoring capability, and non-invasive nature."

p2.l75 -->""However, it exhibited relatively low cellular uptake at commonly used concentrations, which may be partly due to its low fluorescence quantum yield and accumulation in subcellular organelles other than mitochondria."

p2l.72 - comma at the beginning of line

p.3.l84 -->"Because it has a high quantum yield QY of 1.84% in the range of 800–1400 nm and high spatial resolution (SBR=3.84), accurate observation of small metastatic tumors (0.3mm x 1.0mm) can be achieved through NIR-IIa fluorescence imaging with high spatial resolution and precision."

p3l94 --> "This is the first fluorogenic AIE-based probe that can sense lectins in the NIR region, but it suffers from poor water solubility and inability to perform fluorescence imaging of cancer cells."

p4l120 -->"However, DPP-DEG-P with a hydrophilic chain showed better biocompatibility and was able to monitor the fluctuation of ONOO- in APAP-treated hepatocytes with S/N=20 using ratiometric fluorescence imaging."

p13l.390 -->We believe that the development of biological imaging and disease treatment based on DPP derivatives will move in a positive direction, and we hope that this review can encourage more researchers to invest in this field. We also hope to provide some help for the treatment of cancer. Cancer is a daunting challenge, but we believe that with the collaborative efforts of researchers, this problem will be overcome.

Reviewer 2 Report

The manuscript is a collection of data from various resources. Sometimes, not edited well as there are many extra and loosely written sentences. It is long but not enriching the community in a very significant way. However, I recommend reconsider after major revision in form of modification based on my following concerns.  

Line 29: comma missing

Line 37: Check non-scientific writing

Line41: Poor english

Line 42: Poor english

Massive editing could improve the quality of the manuscript. For example, there is no need of Line 132 and 133.  

Line 262: What is polymer molecule? is it polymer or molecule?

Sudden uppercases have been used: For example Line 291

Misleading sentences: Line 304-306

Misleading sentences: Line 312

"water-based" condition?

Line 376-377: Not scientific writing

 "joint efforts of researchers" and?

The conclusion is not meaningful. It basically says the limits but no highlights on the methods to overcome the recent research problems.

Cite recent articles like, J. Mater. Chem. C, 2019,7, 13020-13031

Please check the common comment on English of this paper

Round 2

Reviewer 1 Report

The revised version of the manuscript "Recent Advances of Diketopyrrolopyrrole Derivatives in Cancer Therapy and Imaging Applications" is significatnly better than the original version. There are some minor typos to be corrected:

p2. l.52 :ttumorsthrough

p2. l. 53. "cells. And PDT works primarily"

p10 l. 286 "indicated DBT NPs showed"

p14. l408 the sentence should read probably like this: "Firstly, traditional PDT is highly dependent on oxygen, but the tumor microenvironment is highly hypoxic."

In the conclusion sections the points 1-6 should be uniformly in passive voice.

Reviewer 2 Report

Based on the earlier comments, the manuscript has been revised massively. I recommend accept.
